# Factors Associated with Impact of Event Scores Among Ontario Education Workers During the COVID-19 Pandemic

**DOI:** 10.3390/ijerph21111448

**Published:** 2024-10-31

**Authors:** Iris Gutmanis, Brenda L. Coleman, Robert G. Maunder, Kailey Fischer, Veronica Zhu, Allison McGeer

**Affiliations:** 1Sinai Health, 600 University Ave, Toronto, ON M5G 1X5, Canada; 2School of Public Health, University of Toronto, Toronto, ON M5T 3M7, Canada; 3Department of Psychiatry, University of Toronto, Toronto, ON M5T 1R8, Canada; 4Laboratory Medicine and Pathobiology, School of Public Health, University of Toronto, Toronto, ON M5S 1A8, Canada

**Keywords:** impact of event, traumatic stress, education worker, masking, physical distancing

## Abstract

There is limited information regarding factors related to education workers’ responses to traumatic stress during the COVID-19 pandemic. The study goal was to determine whether personal factors, behaviours that mitigate viral spread, and work-related factors were associated with post-traumatic symptoms. This observational study, embedded within a cohort study, recruited Ontario education workers from February 2021 to June 2023. Exposure data were collected at enrollment and updated annually. Participants completed the Impact of Event Scale (IES) at withdrawal/study completion. Modified Poisson regression was used to build hierarchical models of dichotomized IES scores (≥26: moderate/severe post-traumatic symptoms). Of the 1518 education workers who submitted an IES between September 2022 and December 2023, the incidence rate ratio of IES scores ≥26 was significantly higher among participants who usually/always wore a mask at work (1.48; 95% confidence interval 1.23, 1.79), usually/always practiced physical distancing (1.31; 1.06, 1.62), lived in larger households (1.06; 1.01, 1.12), and reported poor/fair/good health (1.27; 1.11, 1.46). However, models accounted for little of the variance in IES scores, suggesting the need for future studies to collect data on other factors associated with the development of PTSD, such as pre-existing mental health challenges. Early identification of those experiencing traumatic stress and the implementation of stress reduction strategies are needed to ensure the ongoing health of education workers.

## 1. Introduction

Even prior to the World Health Organization’s (WHO) declaration of the COVID-19 pandemic [1], stress and burnout were identified as common health-related issues for education workers [2,3]. Stress associated with teaching has been estimated as comparable to that experienced by ambulance workers, police officers, and social service workers [4]. Post-traumatic stress disorder (PTSD) may develop after witnessing or experiencing what is perceived as a traumatic event [5]. Those living with PTSD experience intense distressing thoughts and feelings related to their experience that last for at least a month [6], but the course can vary [7]. While some people recover within months, others have thoughts and feelings associated with their experience that last much longer [8]. These thoughts/feelings have been grouped into four categories: intrusion (i.e., flashbacks, upsetting dreams), avoidance (i.e., avoiding situations/people/places that may trigger disturbing memories), hyperarousal/reactivity (i.e., overly watchful or reckless behaviour), and alterations in mood and cognition (i.e., ongoing fear, anger, or inability to remember features of the traumatic event) [8].

Several cross-sectional studies, most conducted early in the pandemic, looked for risk factors associated with measures of PTSD symptoms among various populations [9,10,11,12,13,14]. However, direct comparisons are problematic as studies sometimes did not specify, but rather implied, the traumatic event (previous event/COVID-19 pandemic/personal experience of the COVID-19 pandemic), used different instruments to measure PTSD symptoms, and used various scoring methods [15]. These studies found that females are more likely to report symptoms of PTSD [9,10,11,16], but results have not been consistent [12]. Personal and work-related factors associated with fewer symptoms of PTSD include older age [9,10,11], greater number of years of teaching [13], and teaching in a private (versus public) school [12]. Factors associated with increased PTSD symptomatology include lower education level [10,14], lower income [14], having a child [10], history of a psychiatric illness [10], psychiatric drug use of at least three months’ duration [10], and increased alcohol or cannabis use in the past month [14].

Risk factors associated with PTSD symptoms varied over time in an American longitudinal study of adults who were interviewed annually from 2020 through 2022 [16] using the Primary Care-PTSD-4 [17]. These researchers found that the prevalence of probable PTSD decreased between 2020 and 2022 and that PTSD symptomatology was higher in women and in people with larger households (2020 only), those with a history of COVID-19 infection (all time points), and participants who had received at least one (versus no) COVID-19 vaccine (2022 only). Compared with respondents 18–39 years of age, those aged 40–59 years had greater odds of reporting PTSD symptoms, while those 60 years of age or older had lower odds.

In a pre-pandemic report, Lawson et al. reported that interactions with students who had experienced trauma were associated with educator traumatic distress [18]. During the COVID-19 pandemic, school staff not only dealt with their own personal, work-, and family-related issues but may have had to contend with secondary traumatic stress through interactions with students or their family members who experienced COVID-19-related trauma (e.g., death of a family member) as well. The present study builds on the existing COVID-19-related PTSD risk factor literature established in the general population because, to our knowledge, nothing has been published for educators. Yet, it is vital to study the impact of the pandemic on these essential workers.

The goal of this study was to explore factors associated with the intensity of post-traumatic symptoms among education workers. Specifically, this observational study assessed associations between personal factors, preventive behaviours, and work-related factors and PTSD symptoms as measured with the Impact of Event Scale (IES) [19] among Ontario education workers who worked during the COVID-19 pandemic.

## 2. Materials and Methods

### 2.1. Study Design and Sample

Education workers were enrolled in the 34-month prospective cohort study entitled “Study of the epidemiology of COVID-19 in teachers and education workers in elementary and secondary schools in Ontario” from 18 February 2021 until 1 June 2023. The focus of the study was to better understand COVID-19 infections and disease transmission. It recruited education workers who were 18–74 years of age, employed in an Ontario school or school board, worked ≥8 h per week, and planned to continue working for at least three months (for study details, see Coleman et al. [20]). Consenting participants completed baseline surveys at enrollment and annually in September, as well as open surveys of COVID-19 vaccinations and respiratory illness, as needed. The IES was completed only once, within two weeks of study withdrawal (beginning in September 2022) or at study closure (22 December 2023). The study was conducted in accordance with the Declaration of Helsinki and approved by the Sinai Health Research Ethics Board (REB: 20-0343-A; 26 January 2021).

This cross-sectional sub-study used data from the prospective cohort study. Participants with any missing IES data were excluded; no data were imputed. Those who indicated their gender as neither male nor female were excluded from the analysis due to the small sample size (n = 4).

#### Background Information About the Province’s Response to COVID-19

In Ontario, schools were closed four times over the course of the pandemic, twice during the study period that included 15 April to 6 September 2021 and 17 December 2021 to 16 January 2022 [21]. When students and education workers returned to their classrooms/workplaces, mandated viral transmission mitigation strategies, such as masking, were implemented. For the 2020/21 school year, students in grades 4 to 12 were required to wear masks while indoors on school property, while students in kindergarten to grade 3 were encouraged, but not required, to wear masks in indoor spaces. School-based staff who were regularly in close contact with students were provided with medical masks and eye protection as well as other appropriate personal protective equipment (e.g., masks with clear sections if leading classes with students who were deaf) [22]. In 2021/22, students in grades 1 through 12 were required to wear a mask when in hallways, during classes, on school vehicles, or during indoor physical activity. Students were not required to wear masks outdoors. School staff and adult visitors entering the school were also required to wear a mask [23]. Then, as of 21 March 2022, students, staff, and visitors were no longer required to wear a mask at school or on student transportation [24]. However, individuals who were more comfortable wearing a mask were able to continue to do so.

As part of a multi-pronged strategy to reduce illness and COVID-19 transmission among educators and students that included improvements in school ventilation systems, education workers were also asked to promote physical distancing between students and staff and between staff members [25]. Unnecessary furniture was moved, and visual cues, such as tape on the floors and signs on classroom walls, were put in place. Hand sanitizers were also made available in classrooms.

### 2.2. Survey Instruments

#### 2.2.1. Dependent Variables

The IES, a scale with well-established psychometric properties [26], asks participants to indicate the frequency with which they experienced 15 responses associated with traumatic events over the last seven days (scored as: 0: not at all; 1: rarely; 3: sometimes; 5: often). To ensure that respondents were providing answers specific to their pandemic experience, the survey was introduced with “*You have been working throughout the COVID-19 pandemic*”. The first item was changed from “*I thought about it…”* to “*I thought about COVID-19…*”. Similarly adapted IES scales have been used in other studies. For example, Vanaken et al. changed the introduction to the Dutch version of the IES from “*Below is a list of comments made by people after stressful life events*” to “*Please find below a list of statements regarding the situation related to the corona virus (COVID-19)*” [27]. In addition, they modified four of the fifteen items that asked about a past response as the COVID-19 pandemic was ongoing (e.g., “*I was aware that I still had a lot of feeling about it but I didn’t deal with them*” was changed to “*I was aware that I had a lot of feeling about it, but I didn’t deal with them*”). Confirmatory factor analysis found that the structure was similar to that of the original survey, test–retest reliability was adequate, Cronbach’s alpha was 0.75, and the adapted measure was significantly correlated with measures of anxiety and depression. Although the psychometric properties of this study’s adaptations are unknown, it is unlikely that these minor changes had a significant impact on its reliability or validity.

Overall and subscale scores (score range; overall: 0–75; avoidance: 0–40; intrusion: 0–35) are summed item scores. Total IES scores were interpreted using criteria suggested by Hutchings and Devilly [28] and Maunder et al. [29] (0–8: normal/subclinical; 9–25: mild; 26–43: moderate; ≥44: severe symptoms). Subscale scores were dichotomized in alignment with the overall proportion of education workers who had moderate/severe PTSD symptoms (i.e., at the two-thirds/one-third mark), with intrusion scores of ≥13 and avoidance scores of ≥15 considered indicative of a moderate/severe subscale response.

#### 2.2.2. Independent Variables

Information related to age, gender, postal district (five regions based on the first letter of the postal code of residence), education—“What is the highest level of education that you have ever attained?”, subjective health—“In general, would you say your health is…”, and use of anti-anxiety, antidepressants, and anti-insomnia medications—“Do you take any other prescription medications (not including birth control)” were taken from the baseline survey completed at enrollment. Subjective health categories were collapsed due to small cell sizes with *poor* and *fair* responses collapsed with *good*; *very good* and *excellent* responses were also collapsed into one category.

Other personal factors, ones that may have changed over the course of the study, were obtained from the baseline survey completed closest to the IES submission. These included household size—“How many people (other than yourself) live in your home? This includes people who sleep there at least 3 nights per week most weeks of the year” and chronic illnesses—“Have you been diagnosed by a health professional with… each of the following asked as separate questions… asthma, COPD or other chronic lung condition, diabetes, heart disease, cancer that has been treated in the past 5 years, liver or kidney disease, HIV/AIDS or other immune suppressing disease, chronic neurological disorder, and other long-term or chronic condition that has lasted –or is expected to last– at least 6 months”.

Baseline questions specific to preventive behaviours were “While at work (since the beginning of the school year), how often do you wear a mask in others’ presence”; “How often do you wear a mask when you are outdoors, where physical distancing cannot be maintained”; and “While at work (since the beginning of the school year), how often do you physically distance from others?”.

Baseline questions specific to work-related factors included the following: occupation—“What is your current occupation? If you work at more than one, choose the one for which you work more hours”; hours worked—“How many hours per week do you work, on average (paid and unpaid)?”; number of students in close contact—“In an average week, with how many different students do you have close, extended contact (within 2 metres and for 2 min or longer)?”; and highest level of student contact—“Please describe your current level of work-related contact with … students younger than 8 years old, students 8–11 years old, students 12–17 years old”. The highest level of contact with any age group included no close contact or being in the same room/space but rarely within 2 m; being in the same room/space and often within 2 m; being in the same room/space and having physical contact. Changed work hours were defined as the largest difference in the number of hours worked between any two baseline surveys. If the needed information was not in the most recent baseline survey, information was taken from the preceding survey.

The number of COVID-19 infections was established using the illness survey data file, with the total number reported preceding the date of the IES submission used for analysis. Similarly, the number of COVID-19 vaccinations preceding the IES submission was gleaned from the vaccination data file.

### 2.3. Statistical Analysis

Frequency distributions were generated for categorical variables and measures of central tendency for all continuous variables. Modified Poisson regression [30] was used to quantify the exploratory relationship between study independent variables and dichotomized IES scores (≥26: moderate/severe PTSD symptoms). The model was built hierarchically, in three stages. First, backward elimination was used to select personal factors (subjective health, prior COVID-19 infection, household size, any chronic illness, and use of anti-anxiety, antidepression, and anti-insomnia medications), keeping only variables with *p* ≤ 0.2 [31]. Potential confounders (age, gender, postal district, and education) were then added to the reduced model and eliminated if the *p*-value was >0.2. Second, backward elimination of variables included in the (first) personal factors model as well as preventive behaviours (mask at work, mask outdoors, vaccination history, physical distancing at work) was conducted. Third, work-related variables (hours worked, changed work hours, occupation, level of contact with students, and school year) were added to the resulting second model with backward elimination being used to define the final model, again keeping only variables with *p* ≤ 0.2. The final model was assessed for goodness-of-fit and multi-collinearity. Modified Poisson regression models were also generated to assess the association between dichotomized subscale scores and study independent variables. Model building followed the process outlined above. All data were analyzed using Stata v.18 [32].

## 3. Results

### 3.1. Participant Characteristics

Of the 3818 education workers who participated in the parent study, 1518 (39.8%) submitted an IES between September 2022 and December 2023. As shown in Table 1, 1309 (86.2%) participants were female, 1234 were teachers (81.3%), the mean age was 49.0 years, and 194 (12.8%) were taking anti-anxiety, antidepression, or anti-insomnia medications.

### 3.2. Impact of Event Scale

IES scores varied from 0 to 69, with a mean of 20.8 points. Thirty-six percent of respondents had IES scores ≥ 26 indicative of moderate/severe PTSD symptoms (Table 2). The two items with the highest median score (3) were “I thought about COVID-19 when I didn’t mean to” and “I avoided letting myself get upset when I thought about it or was reminded of it” (Appendix A). The highest inter-item correlation was 0.63 between “I had trouble falling asleep or staying asleep because of pictures or thoughts that came into my mind” and “I had dreams about it”. Cronbach’s alpha for the IES was 0.88. The mean intrusion score was 9.6 (95% confidence interval (CI) 9.2, 10.0), and the mean avoidance score was 11.2 (CI 10.8, 11.6).

### 3.3. Adjusted Regression Models of Impact of Event Scale Scores

As seen in Table 3, of the five personal factors hypothesized to be associated with traumatic stress, only subjective health and household size were retained in the final model after adjusting for other covariates. Of the preventive behaviours, more frequent mask-wearing at work, physical distancing at work, and number of COVID-19 vaccine doses received were significantly associated with higher IES scores. None of the work-related variables were retained. The final model showed no evidence of collinearity but explained very little of the total variance in IES scores.

### 3.4. Models of Intrusion and Avoidance (IES Subscales)

Only one personal factor (subjective health) and two preventive behaviours (frequency of mask-wearing and physical distancing) were retained in the final avoidance model adjusted for age and gender (Appendix A). Models of intrusion adjusted for age and gender were more complex and included two personal factors (subjective health, household size), three preventive behaviours (frequency of mask-wearing, physical distancing, increased number of vaccine doses), and three work-related factors (total number of hours worked, maximum change in hours worked, level of student contact). Again, very little of the total variance was explained by either model (pseudo R^2^: avoidance: 1.1%; intrusion: 6.6%).

## 4. Discussion

The WHO has estimated that new illnesses, such as COVID-19, that may require policies to limit contact with non-household members are emerging at the rate of one per year [33]. Essential workers must stay on the job to ensure that healthcare, food supply, public utilities, education, and other services are available throughout these crises. Since susceptibility to circulating illnesses and impact on illness propagation vary by type of work, it is important to understand the risks and outcomes associated with specific groups [34]. This paper provides critical insights into factors associated with moderate to severe PTSD symptoms as experienced by education workers, a group that is often neglected in studies of infectious diseases.

In this study, in 2022–2023, 36% of Ontario education workers had IES scores indicative of moderate/severe PTSD symptoms. Those who reported higher frequencies of mask wearing and physical distancing had higher IES scores. Early in the pandemic, researchers noted an association between the adoption of preventive practices and emotional distress in cross-sectional surveys. For example, Wachira et al. reported that in 2020, American adults responding to a nationally representative survey who masked, practiced social distancing, and practiced hand hygiene were significantly more likely to report feeling nervous, anxious, or on edge for more than a full day in the past seven days compared with those who practiced two or fewer preventive behaviours [35]. Similarly, in a repeated cross-sectional study that recruited Polish adults in 2020 and 2021 [36], the authors reported that wearing a mask and protective gloves and social distancing were associated with higher Impact of Event Survey-Revised (IES-R) [37] scores on two of the three surveys (March, 2020; October to December, 2020; November to December, 2021). In a longitudinal study of illness anxiety during the pandemic, Church et al. [38] recruited English-speaking American adults aged 18 to 65 years who had accessed a crowdsourcing platform in June 2020 with follow-up data collected in December 2020 to assess perceived vulnerability to COVID-19. Adjusted for Time 1 preventive behaviours, germ aversion at Time 1 predicted increased engagement in preventive behaviours at Time 2. This suggests that illness anxiety leads to an increased frequency of preventive behaviours. Of note, however, the frequency of preventive behaviours at Time 1 predicted increased germ aversion at Time 2, suggesting that while engagement in preventive behaviours may decrease short-term anxiety, such behaviours remain associated with illness anxiety in the long term [38]. This bidirectional relationship may also be impacted by personality traits and other affective states. In 2020, Airaksinen et al. [39] found that higher openness, conscientiousness, and neuroticism were associated with a greater likelihood of wearing a face mask among older Europeans. Only higher conscientiousness was associated with a greater likelihood of physical distancing. These traits, that reflect differences in the effort required to adapt to new circumstances and with taking precautions in general, are also associated with a tendency to experience negative emotions such as anxiety.

The association between physical distancing and emotional distress may follow a different pathway. Using data collected in 2020, Cohn-Schwartz et al. found that older community-dwelling European adults who reported higher frequencies of physical distancing felt lonelier [40], and loneliness is a known risk factor for anxiety and PTSD [41]. Previous research has also shown that among teachers, informal peer support and encouragement mitigates the impact of stressors [42]. With limited collegial connections secondary to physical distancing protocols, education workers may have lost a key component of their social and emotional support network. Given that masking and social distancing are proven infectious disease mitigation strategies [43], further longitudinal research into the complex relationship between preventive behaviours, distress processes, and the development of PTSD symptoms is needed.

Ontario education workers reporting excellent/very good health had lower IES scores than workers with poorer subjective health. The association between lower levels of subjective health and increased PTSD symptoms was also noted by Heid et al. [44]. These researchers found that subjective health was associated with PTSD symptoms as measured with the Post-traumatic Stress Disorder Symptom Scale—Self Report among Americans 50 years or older who had lived through Hurricane Sandy. However, in a study conducted in 2022, Greenblatt-Kimron et al. did not find an association between self-rated health and PTSD among older Israelis who had experienced a traumatic event [45]. In the current study, lower self-reported health was also significantly associated with higher scores for both the avoidance and intrusion subscales.

Participants with larger households were more likely to report IES scores indicative of moderate/severe PTSD symptomatology. This is in contrast to Seyahi et al., who found no significant relationship between household size and IES-R scores in a 2020 study of Turkish secondary school teachers [10]. Further, in a study among American adults during the COVID-19 pandemic, the relationship between household size and PTSD, as assessed with the four-item PTSD checklist (Primary Care-PTSD-4) [17], was found to be significant in 2020 only; not in 2021, 2023, or overall [16]. These conflicting results may be due to a more nuanced and possibly time-related relationship between household size and PTSD symptoms, a relationship that may be mediated by factors that impact the quality of familial interactions. For example, Zaken et al. reported higher PTSD-like symptoms among participants with financial issues [46], while Gadermann et al. found that parents with pandemic-induced financial stress reported more conflicts with their children [47], both of which may be exacerbated within larger households.

In contrast to other studies [12,13], we found that work-related factors were not associated with PTSD symptoms. However, in the subscale analyses, more hours worked per week was associated with higher scores on the intrusion scale. Previous research has confirmed that workload intensification is associated with decreased well-being among teachers [13,42]. Of note, when IES data were collected, COVID-19 vaccines were readily available, education workers had returned to in-person teaching, and the WHO had lifted the Public Health Emergency of International Concern declaration [1]. However, a return to in-person teaching may have been associated with increased work hours and increased stress, perhaps due to increases in disruptive student behaviours [48] and the provision of support for students who had fallen behind [49].

In our study, very little of the total variance in IES scores was explained in adjusted models that included personal factors, preventive behaviours, and work-related factors. Roberts et al. [50] also found that very little variance was explained between IES-R scores among physicians living in Great Britain or Ireland during the COVID-19 pandemic: the R^2^ associated with any one personal factor never exceeded 0.1, suggesting that there are many unknown factors and complex pathways leading to the development of post-traumatic symptoms. Future studies are needed so that those most likely to be experiencing traumatic stress are quickly identified and provided with evidence-based supports.

Several study limitations need to be considered when interpreting these observational study results. All study participants were self-selected; some withdrew over time, and only 39.8% submitted an IES, potentially reducing generalizability. Generalizability may also be limited as the timing and intensity of mitigation strategies aimed at reducing viral spread within school systems varied across Canadian provinces. Further, all results are self-reported. As some participants may have been reluctant to disclose symptoms associated with PTSD, study results may have been impacted by social desirability bias. While we have no data from before or early in the pandemic, limiting the ability to identify factors associated with the intensity of PTSD symptoms experienced over the full course of the pandemic, this study followed education workers until the end of 2023, months after the WHO declared the end to the COVID-19 global health emergency. Thus, it provides information on factors associated with longer-lasting PTSD symptoms. As well, additional variance in IES scores may be explained by unmeasured factors that have been identified in the literature, such as pre-existing psychiatric conditions [10], personality [51], and availability of social support [52]. Future studies that include these factors and assess the evolution of PTSD symptoms over the course of a long-lasting traumatic event may shed additional light on who needs additional support and optimal times for their introduction.

## 5. Conclusions

Among Ontario education workers, 36% of respondents assessed between September 2022 and December 2023 were experiencing moderate/severe PTSD symptoms. IES scores were lower for participants with higher levels of subjective health but were higher for those with larger household sizes and with increased frequency of masking and physical distancing at work. However, models accounted for little of the variance in IES scores, indicating the absence of important predictors. Early identification of individuals experiencing traumatic stress followed by the introduction of stress reduction strategies is vital to ensuring the ongoing health of these essential workers.

## Figures and Tables

**Table 1 ijerph-21-01448-t001:** Ontario education worker characteristics; Impact of Event sub-study (September 2022–22 December 2023). Number (%) unless otherwise stated.

Variable	Overall (n = 1518)
Personal factors	
Subjective health	
Poor/fair/good	450 (29.6%)
Very good/excellent	1068 (70.4%)
COVID-19 infections (at time of IES)	
0	399 (26.3)
1	807 (53.2)
≥2	312 (20.6)
Household size	
Mean (95% CI)	3.2 (3.2, 3.3)
Chronic illnesses ^1^	
0	1130 (74.4)
≥1	388 (25.5)
Anti-anxiety, antidepression or anti-insomnia medications	
No	1324 (87.2)
Yes	194 (12.8)
Potential confounding variables	
Age at enrolment (years)	
Mean (95% CI)	49.0 (48.0, 49.0)
Gender	
Female	1309 (86.2%)
Male	209 (13.7%)
Postal district	
Central Ontario	544 (35.8)
Eastern Ontario	286 (18.8)
Northern Ontario	70 (4.6)
Southwestern Ontario	326 (21.5)
Metropolitan Toronto	292 (19.2)
Education	
College diploma or less	110 (7.2)
Bachelor’s degree	219 (14.4)
Teaching certificate	848 (55.9)
Masters/PhD	341 (22.5)
Preventive behaviours	
Wears mask at work	
Never/rarely	910 (60.0)
Occasionally	405 (26.7)
Usually/always	203 (13.4)
Wears mask outdoors	
Never/rarely	1307 (86.1)
Occasionally	68 (4.6)
Usually/always	142 (9.4)
Practices physical distancing	
Never/rarely	501 (33.0)
Occasionally	746 (49.1)
Usually/always	271(17.8)
COVID-19 vaccinations received (at time of IES)	
≤1	22 (1.5)
2	87 (5.7)
3	346 (22.8)
4	434 (28.6)
5	348 (22.9)
6 or 7	281 (18.5)
Work-related factors	
Hours worked per week, average	
Mean (95% CI)	40.5 (39.9, 41.1)
Maximum change in hours worked, per week	
Mean (95% CI)	5.4 (5.0, 5.8)
Occupation	
Teacher/instructor	1234 (81.3)
Non-teaching ^2^	284 (18.7)
Number of students in close contact, average/week	
Mean (95% CI)	60.9 (57.7, 64.2)
Highest level of work-related student contact	
None/Same room but >2 m	1189 (79.0)
Same room and <2 m	259 (17.2)
Physical contact	57 (3.8)
Date IES submitted	
1 September 2022 to 31 August 2023	52 (3.4)
September 1 to 22 December 2023	1466 (96.6)

IES: Impact of Event Scale; 95% CI: 95% confidence interval; ^1^ asthma, chronic lung condition, diabetes, heart disease, cancer treated in the past five years, liver or kidney disease, HIV/AIDS or other immune-suppressing condition, chronic neurological disorder, or other long-term chronic conditions; ^2^ educational assistant, early childhood educator, office and clerical staff, superintendent, human resources, finance, planner, information technology, audio–visual, bus driver, custodian, building maintenance, cafeteria/lunchroom staff, psychologist, social worker, therapist, librarian, nurse, principal, vice principal.

**Table 2 ijerph-21-01448-t002:** Crude Impact of Event Scale and subscale scores for Ontario education workers, September 2022–December 2023, N (%) unless otherwise stated.

Impact of Event Scale	Estimate
Total IES score ^1^	
Mean (95% CI)	20.8 (20.0, 21.4)
Median (IQR)	20 (10, 30)
<26 (subclinical/mild symptoms)	972 (64.0%)
≥26 (moderate/severe symptoms)	546 (36.0%)
Intrusion score ^2^	
Mean (95% CI)	9.6 (9.2, 10.0)
Median (IQR)	8 (3, 14)
<13 (subclinical/mild)	1037 (68.3%)
≥13 (moderate/severe)	481 (31.7%)
Avoidance score ^3^	
Mean (95% CI)	11.2 (10.8, 11.6)
Median (IQR)	10 (5, 17)
<15 (subclinical/mild)	1031 (67.9%)
≥15 (moderate/severe)	487 (32.1%)

CI: confidence interval; IQR: interquartile range; IES: Impact of Event Scale; ^1^ possible score range: 0–75; ^2^ possible score range: 0–35; ^3^ possible score range: 0–40.

**Table 3 ijerph-21-01448-t003:** Modified Poisson regression model comparing Impact of Event Scale scores of Ontario education workers with subclinical/mild post-traumatic symptoms (<26) and moderate/severe post-traumatic symptoms (≥26) (September 2022–December 2023). Incidence rate ratio (95% confidence interval).

Variable	Final Model ^1^
Subjective health	
Poor/fair/good	Referent
Very good/excellent	**0.79 (0.69, 0.90)**
Household size	**1.06 (1.01, 1.12)**
Wears mask at work	
Never/rarely	Referent
Occasionally	**1.20 (1.03, 1.41)**
Usually/always	**1.48 (1.23, 1.79)**
Physical distancing at work	
Never/rarely	Referent
Occasionally	**1.21 (1.01, 1.44)**
Usually/always	**1.31 (1.06, 1.62)**
Number of COVID-19 vaccines received	1.05 (0.99, 1.11) *

Bold identifies group significantly different (*p* < 0.05) from referent; * *p* ≤ 0.20; ^1^ estimates adjusted for other variables in column and confounders: age and gender.

## Data Availability

The datasets generated and/or analyzed during the current study are not publicly available due to information that could compromise the privacy of research participants but are available from the corresponding author on reasonable request.

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
