# Peer review of "Factors Associated with Impact of Event Scores Among Ontario Education Workers During the COVID-19 Pandemic"

_ijerph, 2024, doi:10.3390/ijerph21111448_

Round 1

Reviewer 1 Report

Comments and Suggestions for Authors

The study has the good features of (i) a large sample size and (ii) a special population that needs attention. However, there are some issues and the authors may want to improve the quality considering the below comments.

1. The statement, "Several cross-sectional studies, most conducted early in the pandemic, looked for risk factors associated with measures of PTSD symptoms among various populations" needs citations. 

2. Before introducing the goal of the present study in the Introduction, please provide clear information regarding what the present study could add to the literature gap.

3. Did the authors use any methods to ensure that no participants complete the same survey twice? 

4. I am confused by this statement, "Baseline surveys were completed at enrollment and updated annually in September". Does this mean that the authors have collected follow-up measures?

5. I am also confused by this statement, "Participants with incomplete IES  data were excluded from this sub-study; no data were imputed". What do the authors mean "sub-study" here? Do the authors mean that the present manuscript reports part of one large-scale project? If yes, the authors need to provide clear information of the large-scale project to fulfill the transparency. 

6. When introducing the IES, the authors need to cite some studies using the IES on COVID-19 situation. There are some papers describing how the IES modified for COVID-19 situation have good psychometric properties.

7. The section "2.2.2. Independent Variables" is difficult to read. Please separate the variables in different paragraphs for readers to easily read the variable information clearly. Accordingly, the information regarding the coding for these variables needs to be provided.

8. Given that the present sample is large, I wonder why the authors only dichotomize the independent variable of subjective health. I think that the five levels of the subjective health should not be dichotomized but remain what they are for the data analysis. 

Author Response

  1. The statement, "Several cross-sectional studies, most conducted early in the pandemic, looked for risk factors associated with measures of PTSD symptoms among various populations" needs citations. 

We thank the reviewer for noticing this. Citations have been added.

  1. Before introducing the goal of the present study in the Introduction, please provide clear information regarding what the present study could add to the literature gap.

We have moved some of the introductory material to the methods section (section 2.1.1) and added a sentence linking the literature review and the study goal has now been added to the manuscript (line 70-73 of the clean version).

  1. Did the authors use any methods to ensure that no participants complete the same survey twice? 

We did review the data file for duplicates.

  1. I am confused by this statement, "Baseline surveys were completed at enrollment and updated annually in September". Does this mean that the authors have collected follow-up measures?

The cross-sectional study described in the paper uses data from a cohort study that did collect follow-up data. The cohort study is now more fully described with more details specific to the cross-sectional study also included.

  1. I am also confused by this statement, "Participants with incomplete IES data were excluded from this sub-study; no data were imputed". What do the authors mean "sub-study" here? Do the authors mean that the present manuscript reports part of one large-scale project? If yes, the authors need to provide clear information of the large-scale project to fulfill the transparency. 

As described in the response to issue 4, cohort study details are provided first, followed by a description of the cross-sectional study.

  1. When introducing the IES, the authors need to cite some studies using the IES on COVID-19 situation. There are some papers describing how the IES modified for COVID-19 situation have good psychometric properties.

We thank the reviewers for noting this. We have added information on one such study done by Vanaken et al. who adapted the IES to study PTSD symptoms during the COVID-19 pandemic.

  1. The section "2.2.2. Independent Variables" is difficult to read. Please separate the variables in different paragraphs for readers to easily read the variable information clearly. Accordingly, the information regarding the coding for these variables needs to be provided.

The questions used to collect the study independent variables are now provided in the methods section

  1. Given that the present sample is large, I wonder why the authors only dichotomize the independent variable of subjective health. I think that the five levels of the subjective health should not be dichotomized but remain what they are for the data analysis. 

Despite the overall large sample size, there were very few people who indicated that they had poor or fair subjective health (possibly due to a healthy worker effect). As a result, poor and fair were collapsed with good. Only 68 people rated their health as poor or fair making the cell sizes too small to establish stable estimates. We collapsed the excellent and very good categories since the estimates were very similar for the association between self-rated / subjective health and their score on the IES-R but with wider confidence intervals than when combined (v gd = 0.81; CI 0.70, 0.94 vs exc = 0.73; 0.60, 0.88) – recall the estimate for good vs ‘better than good’: 0.79; CI 0.69, 0.90.

Reviewer 2 Report

Comments and Suggestions for Authors

The study is scientifically sound. Aims to determine whether personal factors, behaviours which mitigate viral spread, and work-related factors were associated with posttraumatic symptoms of 1518 Ontario education workers. This observational study, embedded within a cohort study set22 to dec23 finds significantly higher incidence rate ratio of IES scores (≥26) among participants who usually/always wore a mask at work, usually/always practiced physical distancing, lived in larger households, and reported poor/fair/good health. But accounted for little of the variance in IES scores indicating the absence of important predictors. However, is an important study because show that those most likely to be experiencing traumatic stress could be quickly identified to provide evidence-based supports.

The manuscript is clear, relevant for the field and well presented. The testability of the hypothesis and methods is correct, but more information is needed about missing answers and data availability statements to ensure they are adequate.

Author Response

The manuscript is clear, relevant for the field and well presented. The testability of the hypothesis and methods is correct, but more information is needed about missing answers and data availability statements to ensure they are adequate.

This has been clarified (participants with any missing data were excluded”). Data availability statements are provided as per journal requirements. Thank you.

Reviewer 3 Report

Comments and Suggestions for Authors

While the study addresses a gap in the field by focusing on the long-term impact of the COVID-19 pandemic on the mental health of education workers, in particular on the preventive measures they had to take (use of masks, physical distancing), it is not as relevant today.

On the other hand, the study could improve its methodology by better controlling for confounding variables such as previous mental health conditions, personality traits or socio-economic factors that could influence the development of PTSD. In addition, more detailed longitudinal follow-up and the inclusion of qualitative data (e.g. interviews) could provide deeper insights into how specific stressors affected participants.

It would also be interesting to include more recent studies on education workers during the pandemic to further strengthen the document.

While the paper appears to be well documented and offers valuable insights into a unique population of ‘education workers during the COVID-19 pandemic’, there are some considerations that the authors may wish to address:

  • The models only explain a small part of the variance in PTSD symptoms. Authors could discuss this in more detail and suggest potential variables to improve explanatory power, such as pre-existing mental health conditions or socio-economic factors.
  • The methodology could benefit from additional controls for potentially confounding variables such as pre-existing psychiatric conditions, coping strategies or availability of social support, which could influence PTSD.
  • Further discussion on the longitudinal nature of the dataset and how PTSD symptoms evolved during different stages of the pandemic would be valuable.
  • The sample is limited to Ontario education workers, discussing how the findings could be generalised to other populations outside Ontario could add strength to the paper.

Author Response

  1. While the study addresses a gap in the field by focusing on the long-term impact of the COVID-19 pandemic on the mental health of education workers, in particular on the preventive measures they had to take (use of masks, physical distancing), it is not as relevant today.

With respect, we disagree with the reviewer. We believe that the study is as relevant today as it might have been, if published when the WHO had first declared the COVID-19 pandemic a public health emergency. The pandemic continues; WHO has only lifted the “Public Health Emergency of International Concern” designation (https://www.who.int/europe/emergencies/situations/covid-19). Further, there will likely be more pandemics in the future. What we can learn from this pandemic can be used in future, if we attend to the findings.

  1. On the other hand, the study could improve its methodology by better controlling for confounding variables such as previous mental health conditions, personality traits or socio-economic factors that could influence the development of PTSD. In addition, more detailed longitudinal follow-up and the inclusion of qualitative data (e.g. interviews) could provide deeper insights into how specific stressors affected participants.

We acknowledge that previous mental health conditions, personality traits and socio-economic factors influence the development of PTSD. However, as the focus of the parent study was infection and transmission, these variables were not collected. We have added this to the discussion section, suggesting that these variables should be measured in future studies.

  1. It would also be interesting to include more recent studies on education workers during the pandemic to further strengthen the document.

The authors did look for other publications regarding education workers but found only those referenced. They are not a group of essential workers that are studied often.

  1. While the paper appears to be well documented and offers valuable insights into a unique population of ‘education workers during the COVID-19 pandemic’, there are some considerations that the authors may wish to address:
  • The models only explain a small part of the variance in PTSD symptoms. Authors could discuss this in more detail and suggest potential variables to improve explanatory power, such as pre-existing mental health conditions or socio-economic factors.

We agree with the reviewer and have added this to the limitations section of the paper.

  • The methodology could benefit from additional controls for potentially confounding variables such as pre-existing psychiatric conditions, coping strategies or availability of social support, which could influence PTSD.

We agree with the reviewer, and have added this to the limitations section of the paper

  • Further discussion on the longitudinal nature of the dataset and how PTSD symptoms evolved during different stages of the pandemic would be valuable.

Understanding the evolution of PTSD during the COVID-19 pandemic was beyond the scope of the study. We agree that a future longitudinal study should be conducted.

  • The sample is limited to Ontario education workers, discussing how the findings could be generalised to other populations outside Ontario could add strength to the paper.

Further discussion factors that may limit the generalizability of the study findings has now been included, thank you. 

Thank you for your helpful suggestions.

Round 2

Reviewer 3 Report

Comments and Suggestions for Authors

I don't have comments for authors. They addressed each comment that I wrote them for revision.